# Randomised feasibility trial into the effects of low-frequency electrical muscle stimulation in advanced heart failure patients

Stuart Ennis,[1,2] Gordon McGregor,[1,3] Thomas Hamborg,[4] Helen Jones,[5] Robert Shave,[2] Sally J Singh,[3,6] Prithwish Banerjee[1,3]

[1]Department of Cardiac Rehabilitation, University Hospitals Coventry & Warwickshire NHS Trust, Coventry, UK
[2]School of Sport, Cardiff Metropolitan University, Cardiff, UK
[3]Centre for Applied Biological and Exercise Sciences & Centre for Technology Enabled Health Research, University of Coventry, Coventry, UK
[4]Clinical Trials Unit, University of Warwick, Coventry, UK
[5]Sport and Exercise Sciences, Liverpool John Moores University, Liverpool, UK
[6]Centre for Exercise and Rehabilitation Science, University Hospitals of Leicester NHS trust, Leicester, UK

**Correspondence to**
Stuart Ennis;
stuart.ennis@uhcw.nhs.uk

## ABSTRACT

**Objectives** Low-frequency electrical muscle stimulation (LF-EMS) may have the potential to reduce breathlessness and increase exercise capacity in the chronic heart failure population who struggle to adhere to conventional exercise. The study's aim was to establish if a randomised controlled trial of LF-EMS was feasible.

**Design and setting** Double blind (participants, outcome assessors), randomised study in a secondary care outpatient cardiac rehabilitation programme.

**Participants** Patients with severe heart failure (New York Heart Association class III–IV) having left ventricular ejection fraction <40% documented by echocardiography were eligible.

**Interventions** Participants were randomised (remotely by computer) to 8 weeks (5×60 mins per week) of either LF-EMS intervention (4 Hz, continuous, n=30) or sham placebo (skin level stimulation only, n=30) of the quadriceps and hamstrings muscles. Participants used the LF-EMS straps at home and were supervised weekly

**Outcome measures** Recruitment, adherence and tolerability to the intervention were measured during the trial as well as physiological outcomes (primary outcome: 6 min walk, secondary outcomes: quadriceps strength, quality of life and physical activity).

**Results** Sixty of 171 eligible participants (35.08%) were recruited to the trial. 12 (20%) of the 60 patients (4 LF-EMS and 8 sham) withdrew. Forty-one patients (68.3%), adhered to the protocol for at least 70% of the sessions. The physiological measures indicated no significant differences between groups in 6 min walk distance(p=0.13) and quality of life (p=0.55) although both outcomes improved more with LF-EMS.

**Conclusion** Patients with severe heart failure can be recruited to and tolerate LF-EMS studies. A larger randomised controlled trial (RCT) in the advanced heart failure population is technically feasible, although adherence to follow-up would be challenging. The preliminary improvements in exercise capacity and quality of life were minimal and this should be considered if planning a larger trial.

**Trial registration number** ISRCTN16749049

## Strengths and limitations of this study

► To our knowledge, this was the first study to evaluate the design of a study into LF-EMS in patients with advanced (New York Heart Association class III–IV) heart failure.

► Analysis of recruitment, retention and adherence in this hard to reach group contributes useful knowledge to the heart failure literature on how practical exercise interventions could be implemented.

► This study was a real-world feasibility study. Patients with advanced heart failure were recruited when deemed eligible by experienced clinicians based on available information. This approach can be subjective and lead to variability in disease severity in our sample. However, this is in keeping with the pragmatic aim of our trial and provides external validity to our findings.

► This study had a small sample size and was not powered or designed to assess the effects of LF-EMS in advanced heart failure. The findings should therefore be considered preliminary.

## INTRODUCTION

Chronic heart failure (CHF) affects approximately 26 million people worldwide[1] and is associated with a poor prognosis; 30%–40% of patients diagnosed with heart failure die within a year.[2] Patients in New York Heart Association (NYHA) class III/IV are unable to perform the simplest daily activities, become depressed and have a poor quality of life.[3]

Regular aerobic exercise reduces breathlessness and muscle dysfunction for individuals with CHF while improving exercise capacity.[4–6] According to the ExTraMATCH meta-analysis,[7] exercise training leads to a 35% relative reduction in mortality, similar to the effects of beta-blockers[8] and angiotensin-converting-enzyme inhibitors.[9] However, those with advanced CHF are often so limited that they are unable to gain the holistic benefits of exercise.[4 7]

Electrical muscle stimulation (EMS) may provide an alternative rehabilitative therapy for this group. In patients with mild to moderate CHF, EMS can improve muscle strength of the legs, exercise capacity and quality of life.[10–12] Low-frequency (4–5 Hz) EMS (LF-EMS) produces shivering-like subtetanic muscle contractions that can stimulate an aerobic response equivalent to 51% of maximal oxygen uptake.[13] Therapeutic levels of aerobic exercise can thus be achieved passively by LF-EMS,[14] and it has been shown to be comfortable and well tolerated in healthy individuals and those with mild to moderate CHF.[15 16] However, the impact of LF-EMS in patients with advanced heart failure (New York Heart Association (NYHA) class III/IV) is currently unknown. As patients with advanced heart failure have shown poor uptake and adherence to intervention studies,[17] a preliminary study was needed to determine the feasibility of LF-EMS in this patient cohort prior to the development of a large-scale definitive trial.

Based on recommendations for good practice in the design of pilot and feasibility studies,[18] this study was undertaken with the following aims: to (1) test the robustness of the study protocol for a potential future trial; (2) estimate rates of recruitment, consent and retention; (3) determine the tolerability of the LF-EMS intervention and the effectiveness of the sham placebo in the NYHA III/IV CHF population and (4) gain initial estimates of the efficacy of LF-EMS for all potential primary outcomes. This can be used for sample size calculations in future substantive trials.

## METHODS
### Experimental design
This feasibility study used a double-blind parallel group randomised control design. Participants were randomised to either LF-EMS or 'sham' placebo for a period of 8 weeks and blinded to group allocation. Outcomes were assessed at baseline (prerandomisation), 8 week and 20 week follow-up.

### Recruitment and screening
Between October 2013 and March 2015, University Hospital Coventry and Warwickshire, (UHCW) Hospital NHS Trust heart failure clinics lists were screened for patients fulfilling the eligibility criteria for the study. Sixty eligible participants were recruited. The study conformed to the Declaration of Helsinki and was approved by the local NHS Ethics Committee. All participants provided written informed consent.

### Randomisation
The trial statistician, in conjunction with Warwick Clinical Trials Unit, generated the randomisation sequence remotely (by computer) using permuted block randomisation. Group allocation was concealed from outcomes assessors and participants.

### Participants
Male and female adults, >18 years old, with stable CHF, documented by echocardiography of left ventricular systolic dysfunction (ejection fraction <40%) were eligible for the study. All participants had NYHA functional class III–IV symptoms as judged by an experienced heart failure cardiologist. Participants were required to be medically stable, defined as the absence of hospital admission or alterations in medical therapy within the preceding 2 weeks. Exclusion criteria for safety and practical reasons were: (1) presence of implantable cardiac devices, (2) serious cardiac arrhythmias, (3) neurological disorders or previous stroke significant enough to limit exercise, (4) orthopaedic problems that prevented walking, (5) neuromuscular disease, (6) dementia or (7) a mid-thigh circumference of >50 cm (due to the size of the LF-EMS straps).

### LF-EMS stimulation
The LF-EMS equipment (Biomedical Research Limited, Galway, Ireland) consisted of a pair of neoprene straps containing built-in adhesive gel electrodes. The equipment is Conformite Europeene (CE) marked under the European Medical Device Directive. The stimulator current waveform was designed to produce rhythmical contractions in the leg muscle groups occurring at a pulse frequency of 4–5 Hz (pulse width: 620 µs). The maximum peak output pulse current used was 140 mA.

### LF-EMS intervention
Participants used the LF-EMS or sham placebo for 1 hour, five times a week, for eight consecutive weeks. Of the five hourly sessions per week, four were completed unsupervised in the participant's own home. The remaining session was conducted in a cardiac rehabilitation outpatient setting under the supervision of an exercise physiologist. The LF-EMS technology was retrospectively interrogated (ie, at the weekly supervised sessions) to report date, frequency, duration and stimulation intensity.

### 'Sham' placebo intervention
In the sham arm of the study, participants were provided with identical straps and electrodes. In contrast to the LF-EMS group, the controller was programmed to deliver a very low level of stimulation (frequency: 99 Hz, pulse width: 150 µs, maximum current amplitude: 7.3 mA). This provided sensory input to the skin surface but little or no muscle activation. Participants in the sham group had the same induction, supervision and follow-up as the intervention arm.

### Outcome measures
#### Feasibility criteria
In relation to the design of pilot and feasibility studies, Thabane et al[19] recommend stipulating criteria for success 'a priori'. The feasibility criteria were:
1. Recruitment rate: at least 40% of eligible participants recruited to the trial.

2. Retention: no more than 33% of participants drop out during the intervention period.
3. Adherence: 66% of participants tolerate the intervention and adhere to the protocol for ≥70% of the intervention period.
4. Placebo efficacy: participants would be able to guess their group allocation no more often than would be expected by chance.

### Primary outcome
#### 6 min walk test
The 6 min walk test (6MWT) was conducted in accordance with the American Thoracic Society guidelines.[20] Participants were instructed to walk as far as possible in 6 min along a 30 m, flat, obstacle free corridor, turning 180° at the end of every 30 m. Standardised instructions and verbal encouragement were given.

### Secondary outcomes
#### Isometric muscle strength
A hand-held dynamometer (MicroFET2 Torque/Force indicator, Hoggan Health Industries, Utah, USA) validated for assessing functional leg strength in elderly populations was used.[21] Participants sat in an elevated chair and were instructed to maximally extend the knee while the assessor provided an equal and opposite resistive force, against the lower shin. The mean force generated was measured in Newtons.

### Quality of life: Minnesota Living with Heart Failure Questionnaire
The Minnesota Living with Heart Failure questionnaire (MLHFQ) is a disease validated questionnaire,[22] that has been extensively used in heart failure studies. Questionnaire scores range from 0 to 105, with higher scores reflecting lower quality of life. Participants were asked to answer each question based on their perception of health in the week previous to testing.

### Physical activity levels
Physical activity levels were measured by the Bodymedia© SenseWear Pro3 Armband. The multiplane accelerometer was worn continuously for the 7 days prior to testing to determine total energy expenditure per 24 hours period was used as the main indicator of physical activity.

### LF-EMS acceptability questionnaire
At the end of the trial, participants were given a brief questionnaire used in previous LF-EMS studies,[13 14] to collect feedback on the acceptability of using LF-EMS regularly. Questions used the Likert scale to discern cognitive and affective components of attitudes[23] about ease of use, comfort, tolerability and overall satisfaction.

### Safety: blood test
Venous blood samples were taken at baseline, 4 weeks and 8 weeks to assess creatine kinase (CK), urea and electrolytes. Participants would discontinue the trial if levels exceeded the upper limit of normal reference ranges.

### Data analysis
Data analyses for the feasibility objectives of this study were descriptive, based on the predetermined levels specified above. CI (set at 95%) were calculated for all secondary outcome measures in both groups and paired two-sample t-test conducted for between group comparisons. Intent-to-treat analysis was employed in this study as is recommended for clinical trials.[24]

## RESULTS
### Feasibility criteria outcomes
#### Recruitment
There were 171 eligible participants identified in the Coventry and Warwickshire area from November 2013 to April 2015. Sixty of 171 eligible participants (35.08%) were recruited to the trial. Participants were randomised and started on the trial during this period and were followed up until data collection finished in August 2015. Participant characteristics are presented in table 1.

#### Retention
Twelve of the 60 participants (4 LF-EMS, eight sham) (20%) withdrew and did not finish the intervention period (see figure 1). Of these, only three found the intervention intolerable (one LF-EMS, two sham). Other reasons for dropout were: deterioration in health (n=6), family problems (n=2) and implantation of a cardioverter defibrillator (n=1). Only 22 (45%) of those completing the intervention period returned for follow-up testing at 20 weeks. Reasons for non-follow-up were: deterioration in health (n=9), excluded due to implantation of cardiac resynchronisation therapy device (n=2), declined to take part without further explanation (n=13) and could not be contacted after repeated attempts (n=3).

#### Adherence
Forty-one (85.4%) of the 48 participants (22 LF-EMS and 19 sham) who completed the intervention period (68.3% of the total sample) adhered to the strict protocol for the majority (>70%) of the 8 weeks. Interrogation of the LF-EMS controllers revealed that participants in the LF-EMS group became more tolerant to the intervention; mean stimulation intensity increased from 57.79 mA (95% CI 51.16 to 64.42) during week 1 of the study to 84.86 mA (95% CI 75.44 to 94.28) by week 8, an improvement of 46.5%.

#### 'Sham' Placebo
The sham placebo for the study appeared to be convincing as only 61% of participants guessed their treatment group correctly. The 95% CI for the proportion of participants guessing correctly was (46% to 74%)% and thus not significantly different from 50% which would be expected by chance. Furthermore, participants demonstrated an inclination to guess that they were randomised to LF-EMS regardless of group allocation.

**Table 1** Baseline demographic and clinical characteristics of the LF-EMS and sham placebo groups

| Demographics | LF-EMS (n=30) | Sham (n=30) |
|---|---|---|
| Male (n) | 20 (66%) | 22 (73%) |
| Age (years) | 66.5±7.8 | 66.8±13.5 |
| Body mass index (kg/m$^2$) | 30.1±4.9 | 27.8±4.8 |
| Comorbidities | | |
| Prev MI/PCI/CABG | 17 (56%) | 11 (36%) |
| Diabetes | 12 (40%) | 10 (33%) |
| COPD | 9 (30%) | 8 (26%) |
| AF | 20 (66%) | 16 (53%) |
| Hypertension | 13 (43%) | 10 (33% |
| CKD | 5 (16%) | 13 (43%) |
| Clinical | | |
| NT-pro-BNP (pg/mL) | 3086±3746 | 2046±2545 |
| Creatinine (mmol/L) | 108±49 | 113±39 |
| LVEF % | 39±11* | 22±12† |
| BP$_{sys}$ (mm Hg) | 118±16 | 126±17 |
| BP$_{dia}$ (mm Hg) | 69±9 | 74±14 |
| NYHA III | 24 (80%) | 22 (73%) |
| NYHA IV | 6 (20%) | 8 (26%) |

Data presented as mean±SD or absolute number and percent.
*n=10. Ejection fraction could not be accurately assessed in all patients due to poor body habitus/atrial fibrillation. An experienced cardiac sonographer made an 'eyeball' assessment of poor left ventricular function for all other participants.
†n=5. See previous comments.
AF, atrial fibrillation; BP$_{dia}$ (mm Hg), diastolic blood pressure; BP$_{sys}$ (mm Hg), systolic blood pressure; CABG, coronary artery bypass graft surgery; CKD, chronic kidney disease; COPD, chronic obstructive pulmonary disease; LF-EMS, low-frequency electrical muscle stimulation; LVEF, left ventricular ejection fraction; MI, myocardial infarction; NT-pro-BNP (pg/mL),N-terminal pro B-type natriuretic peptide; NYHA, New York Heart Association; PCI, percutaneous coronary intervention.

### Safety
No abnormalities were detected in CK, urea or electrolytes taken before, during or after the study. Likewise, no adverse events due to the intervention were recorded in either group.

### Primary outcome: 6MWT
Non-significant improvements after LF-EMS (8 week time point) and sham groups were observed in 6 min walk distance (6MWD) with a mean increase from baseline of 24 m (p=0.13) in the LF-EMS group (table 2).

### Secondary outcomes
Table 2 shows the mean values of the secondary outcome measures at each time point. There were no significant differences between groups in the change from baseline for any of the secondary outcome variables (table 3). There was a non-significant improvement in quality of life in both groups.

### Acceptability questionnaire
Participants' responses to the LF-EMS acceptability questionnaire are summarised in table 4. The mean response to putting on the straps was 2 ('quite easy') and the overall mean satisfaction of participants with the intervention was 6 out of 10. Mean responses to comfort, sensation, tolerability and continued use of LF-EMS were between 3 (medium) and 4 (quite hard/unpleasant).

### Sample size calculation
The point estimate from the study and the upper CI limit of this estimate were calculated. The upper CI limit was used for the sample size calculation. For detecting the observed difference of 13.4 m in this study a sample size of 240 patients per group would be required. However, a recent study[25] suggested that the minimal clinically important difference for 6MWD is 36 m in patients with mild–moderate CHF. The clinical benefit of the effect size in this study should be considered before proceeding with a larger trial

## DISCUSSION
The predetermined criteria for proceeding to a larger trial were achieved for dropout (20%), adherence (68.3%) and sham placebo efficacy (61.53% participants guessed correctly). However, only 35.06% of eligible patients were recruited, below the target of 40%. Initial outcome measures revealed no significant difference between intervention and placebo groups, although there was a non-significant improvement in 6MWD and quality of life after LF-EMS.

### Feasibility outcomes
#### Recruitment
Percentage uptake (35.06%) of eligible patients in the study was below the predetermined criteria of 40%. This is similar to the poor uptake of conventional cardiac rehabilitation (CR) nationally in the UK: <40% of eligible patients with heart failure accessed CR in the most recent National Audit of Cardiac Rehabilitation.[26]

#### Retention/adherence/tolerance
One strength of this study is the good level of adherence (68.3%) and retention (80%) compared with other clinical studies; In the Heart Failure: A Controlled Trial Investigating Outcomes in Exercise training (HF-ACTION) trial,[27] only 40% of patients in the exercise group (n=1159) reported adherence to recommended training volumes after 3 months. This may have been because of the ease of independent use at home of LF-EMS, in combination with the weekly supervised sessions with an exercise physiologist. The patients recruited in the present trial were more debilitated yet they engaged more with LF-EMS than those in the HF-ACTION trial,[27] suggesting that LF-EMS maybe more acceptable to this population than conventional exercise.

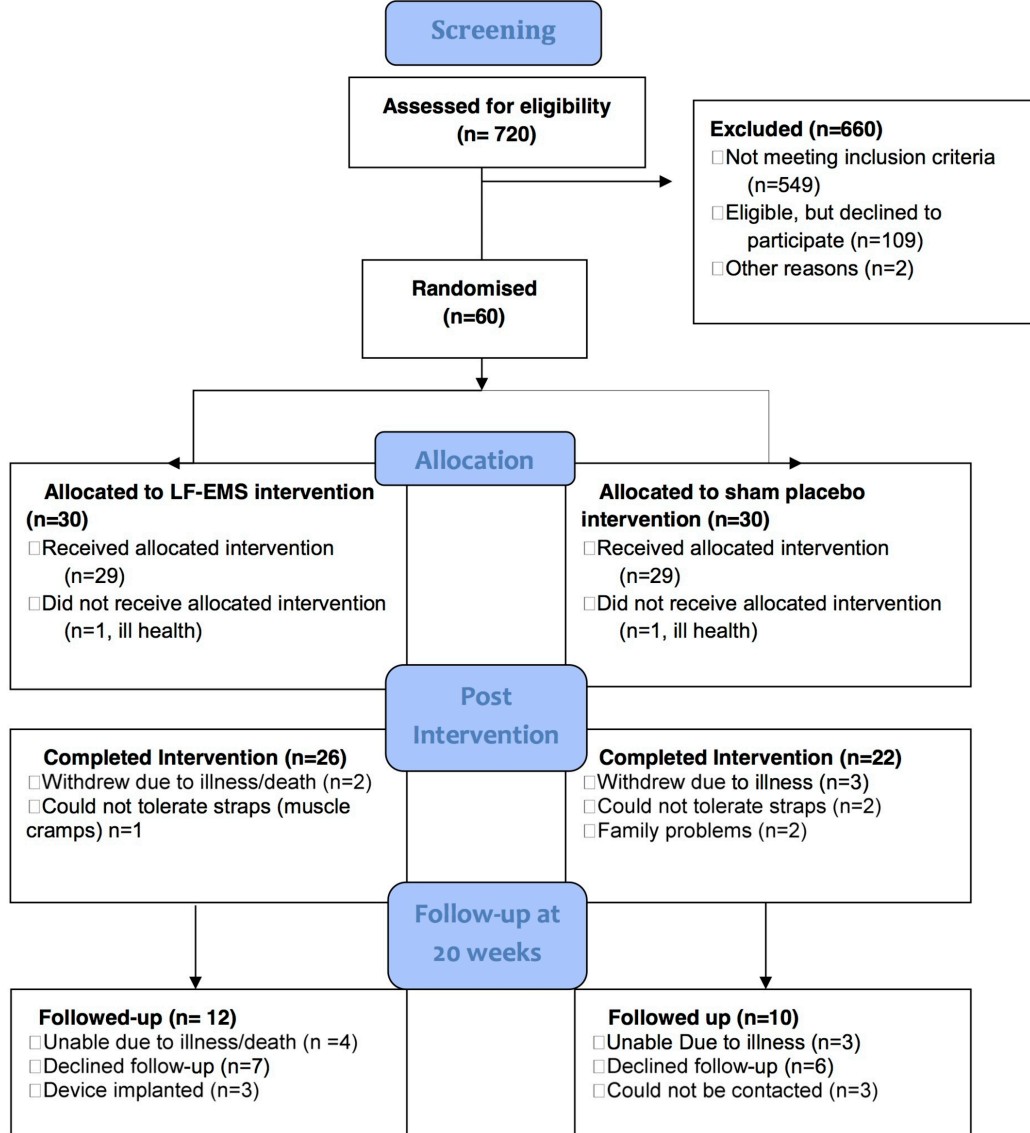

**Figure 1** Flow diagram of a single-centre blinded parallel group randomised feasibility trial of EMS versus sham placebo in patients with severe heart failure. LF-EMS, low-frequency electrical muscle stimulation.

The dropout at 3 months follow-up was lower than expected due to ill health, device implantation and apathy, and would be challenging to overcome in a larger trial. Strategies to combat dropout could include combining assessment with clinical patient appointments to ensure compliance or arranging home visits for some assessments.

Feedback from the acceptability questionnaires may also be useful in curtailing dropout in a larger trial: the LF-EMS group generally thought that wearing the straps for an hour was 'medium' to 'quite hard/unpleasant'. Continued use of a LF-EMS was deemed challenging also so it is possible that a reduced frequency of LF-EMS while still maintaining a sufficient dose, for example, 3×1 hour a week may enhance long-term adherence.

Tolerance to the LF-EMS intervention improved during the study. Mean current intensity increased by 46% from week 1 to 8. This tolerance effect is in keeping with an earlier study by Crognale et al[13] that showed a 20% increase in healthy active adults. The active adults tolerated higher absolute stimulation levels than in this study, both before and after habituation, suggesting that patients with advanced CHF are subjectively less tolerant to LF-EMS than a healthy population. In addition, the user feedback collected seems to support this view. Vivodtzev and colleagues[28] examined factors determining tolerance of EMS in patients with pulmonary disease. The study reported that lower tolerance to EMS was associated with greater severity of condition, fat free mass and inflammatory response. It is possible that the same is true in the CHF population but more research is needed to confirm this.

**Outcome measures**

Baseline 6MWD was higher in our study sample than in other advanced heart failure studies.[29] This may have been due to high variability because of a few outliers

**Table 2**  Outcome measurements: time point averages and 95% CI

| Outcome | Time point | LF-EMS | Sham |
|---|---|---|---|
| Mean 6MWD (m) (95% CI) | Baseline (n) | 283 (237 to 328) 29 | 290 (243 to 337) 29 |
| | 8 weeks (n) | 312 (262 to 362) 26 | 318 (270 to 365) 22 |
| | 20 weeks (n) | 257 (173 to 342) 12 | 226 (126 to 325) 10 |
| Mean leg strength (N) (95% CI) | Baseline (n) | 234.3 (196.5 to 272.) 29 | 297.5 (253 to 342) 29 |
| | 8 weeks (n) | 224.9 (187.5 to 262.3) 25 | 321 (267.8 to 374.3) 22 |
| | 20 weeks (n) | 181.6 (131.7 to 231.5) 11 | 207.1 (148.6 to 265.7) 10 |
| Mean QoL score (95% CI) | Baseline (n) | 53.1 (42.7 to 63.5) 28 | 50 (40 to 60.1) 29 |
| | 8 weeks (n) | 43.9 (34.2 to 53.5) 25 | 43.1 (30.9 to 55.3) 22 |
| | 20 weeks (n) | 51.7 (31.6 to 71.8) 12 | 37.0 (16.9 to 57) 10 |
| Mean TEE (J) (95% CI) | Baseline (n) | 63 438 (56 170 to 70 705) 25 | 65 371 (59 675 to 71 067) 27 |
| | 8 weeks (n) | 59 783 (51 094 to 68 471) 19 | 59 687 (50 630 to 68 745) 17 |
| | 20 weeks (n) | 61 878 (53 345 to 70 410) 7 | 63 541 (55 795 to 71 287) 6 |

6MWD, 6 min walk distance, QoL, quality of life; TEE, total energy expenditure.

in each group. This reflects the subjective nature of the NYHA classification system. However, signs and symptoms of advanced heart failure were primarily the eligibility criteria for this study and not 6MWD. In addition, the ≤300 m distance cut-off (below which our baseline mean falls) is often cited, as prognostically important and reflective of advanced disease in many investigations.[30–32] The non-significant improvements in exercise capacity as measured by 6 min walk were smaller than those in a meta-analysis of EMS in patients with

**Table 3**  Changes from baseline averages and 95% CI

| Outcome | Time point | LF-EMS | Sham | p Value |
|---|---|---|---|---|
| Mean 6MWD (m) (95% CI) | Baseline to 8 weeks (n) | 24 (9 to 40) 26 | 9 (–4 to 22) 22 | 0.1366 |
| | Baseline to 20 weeks (n) | 0 (–32 to 31) 12 | –26.30 (–63 to 11) 10 | 0.2409 |
| Mean leg strength (N) (95% CI) | Baseline to 8 weeks (n) | –9.2 (–28.9 to 10.5) 25 | 6.0 (–19.3 to 31.4) 22 | 0.3244 |
| | Baseline to 20 weeks (n) | –43.4 (–78.7 to –8.2) 11 | –74.1 (–116.3 to –31.9) 10 | 0.2223 |
| Mean QoL score (95% CI) | Baseline to 8 weeks (n) | –7.6 (–15.5 to 0.3) 25 | –4.7 (–10.5 to 1.0) 22 | 0.5505 |
| | Baseline to 20 weeks (n) | 1.5 (–12.5 to 15.7) 12 | –14.0 (–34 to 6) 10 | 0.1610 |
| Mean TEE (J) (95% CI) | Baseline to 8 weeks (n) | –4635 (–3963 to 4692) 19 | –8168 (–14 342 to –1995) 17 | 0.5108 |
| | Baseline to 20 weeks (n) | 1686 (–6435 to 9809) 7 | 4177 (–7695 to 16 050) 6 | 0.6634 |

6MWD, 6 min walk distance; QoL, quality of life; TEE, total energy expenditure.

**Table 4** Mean responses to acceptability questionnaire and SD

| Question | Mean response | |
|---|---|---|
| 1. I found putting on the straps (1-easy, 5-hard) | 2.0 | (±1.17) |
| 2. At the highest intensity I found the comfort level (1-acceptable, 5-unacceptable) | 3.5 | (±1.19) |
| 3. Overall I found the sensation (1-pleasant, 5-unpleasant) | 3.3 | (±1.13) |
| 4. I found putting on the LF-EMS for an hour (1-easy, 5-hard) | 3.1 | (±1.08) |
| 5. I think I would find staying on a LF-EMS training routine (1-easy, 5-hard) | 3.4 | (±1.29) |
| 6. Overall satisfaction with LF-EMS as a way of improving your fitness (1-none,10 extremely satisfied) | 6.0 | (±1.94) |

LF-EMS, low-frequency electrical muscle stimulation.

heart failure by Smart et al.[10] These authors reported a combined improvement in 6MWD of 46.9 m versus usual care or placebo, compared with the effect size of 13.2 m in this study. However, patients in this study were more symptomatic than those included in the meta-analysis,[10] and thus had a lower baseline exercise capacity (286 m vs 342 m). Nevertheless, the mean relative increase (5%) in walk distance of participants in the LF-EMS group is within the measurement error associated with this test[33] and probably should not be considered clinically significant.[25] The extrapolation from these results that patients with severe CHF are beyond help from EMS maybe premature; a longer training period maybe required to show meaningful changes in exercise capacity, particularly as some participants took longer to tolerate meaningful EMS intensities than others.

Quality of life (MLHFQ) improved in both groups after the intervention. This may, in part, relate to the psychosocial benefits of engaging with researchers regularly in the cardiac rehabilitation facility.[34] The placebo effect of both interventions and its influence on patients' perception of well-being should not be underestimated.

Based on previous research by Banerjee et al,[15 16] and numerous high-frequency EMS studies,[12 35 36] improvement in leg strength after use of LF-EMS was expected. The current trial however showed no significant change in muscle strength. Muscle wasting, prevalent in many patients with advanced heart failure,[37] could explain this observation. The chronic impairment of muscle tissue caused by heart failure affects the muscle and skin nerve receptors and hence contractility of the weakened muscle.[38] Participants with more functional leg muscles therefore may have received greater stimulus to muscle tissue that others did for the same level of current intensity. This suggests that LF-EMS may not be effective for all patients with advanced CHF.

## Limitations

The sample for this study was small as is recommended for feasibility studies[19] and this limits the external validity of our findings. Participants were deemed eligible for the study based on the judgement of experienced heart failure clinicians using available knowledge. This may have led to greater variability in disease severity/limitation than was intended. The current amplitude (mA) stimulus intensity that participants chose to use was a limitation to the study design. Participants were instructed to adhere to the 'maximum tolerable intensity' during LF-EMS sessions. Due to considerable individual differences in the subjective perception of discomfort associated with EMS, it is therefore likely that there was variability in the intensity that individuals received.

## CONCLUSION

As some of the predetermined feasibility criteria were met in this trial, a larger study into the effects of LF-EMS on patients with advanced heart failure could be undertaken. However, this 'difficult to engage with' patient group would be very challenging to recruit and follow-up in sufficient numbers to provide definitive data on its efficacy. The improvements seen in this study in 6MWD, and quality of life measures, were not statistically significant. Leg strength and physical activity levels showed no significant change. An intervention period >8 weeks could be considered to give participants more time to adjust to the intervention. More investigation is required to determine which patients with CHF are unresponsive to LF-EMS due to severe muscle dysfunction.

A larger trial may be feasible with this difficult population: however, it is unlikely that the non-significant improvement in exercise capacity and quality of life found in this pilot study justifies a larger pragmatic trial.

**Acknowledgements** The authors would like to thank Matthew Annals for his recruitment expertise and for administering the intervention to participants. They would also like to thank Josie Goodby for her assistance and patience during all assessments.

**Contributors** SE, GM and PB: contributed to the conception of the work. SE, GM, PB, SS, HJ, RS and TH: contributed to the design of the work. SE and GM: contributed to the acquisition, of the work. SE, GM, PB, SJS, HJ, RS and TH: contributed to the, analysis, or interpretation of data for the work. SE and GM: drafted the manuscript. PB, SJS, HJ, RS and TH: critically revised the manuscript. All authors: gave final approval and agree to be accountable for all aspects of work ensuring integrity and accuracy.

**Funding** This study was funded by a National Institute for Health Research (NIHR) Research for Patient Benefit (RfPB) award.

**Disclaimer** This article/paper/report presents independent research funded by the National Institute for Health Research (NIHR). The views expressed are those of the author(s) and not necessarily those of the NHS, the NIHR or the Department of Health.

**Competing interests** None declared.

**Patient consent** no identifiable patient information is contained within the submitted manuscript.

**Ethics approval** NRES Committee West Midlands - Coventry & Warwickshire13/WM/0240.

**Provenance and peer review** Not commissioned; externally peer reviewed.

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
