## [Reviewer comments · BMJ Open]

ARTICLE DETAILS

TITLE (PROVISIONAL)	Randomised feasibility trial into the effects of low frequency electrical muscle stimulation in advanced heart failure patients
AUTHORS	Ennis, Stuart; McGregor, Gordon; Hamborg, Thomas; Jones, Helen; Shave, Rob; Singh, Sally; Banerjee, Prithwish

VERSION 1 - REVIEW

REVIEWER	Patricia Palau Hospital General Universitario de Castellón Cardiology Department Universitat Jaume I
REVIEW RETURNED	24-Mar-2017

GENERAL COMMENTS	In this randomized study authors evaluated the feasibility and effectivity of LF-EMS in patients with heart failure with reduced left ventricular ejection fraction (HFpEF). Authors concluded this study was feasible however it did not show any improvement in functional or QoL in patients with advanced HFrEF. The hypothesis is pertinent and the protocol was meticulous. Authors should be congratulated. The main strength of this finding is that patients with advanced HFrEF are able to tolerate a home-based LF-EMS program. However, there are some issues that deserve to be commented/clarified. 1. Study population: It is necessary to specify the inclusion and exclusion criteria according to recent guidelines.2. An important limitation is that heart failure diagnosis was not based on natriuretic peptides.3. Authors should specify the associated comorbidities, echocardiographic findings and baseline natriuretic peptides in Table 14. Authors selected 60 symptomatic HFrEF (median age 66 years), overweight (BMI 27.8-30.1) with impaired functional capacity by cardiopulmonary exercise testing (CPX) due to shortness of breath or fatigue and NYHA III-IV. However, this is not congruent with the results showed in table 1 where the distance of 6-MWT is 237-337 m. Likewise, previous studies in obese subjects have shown that these individuals display per se a significant impairment of functional capacity. Authors should clarify how many patients exhibit NYHA class II- III –IV and the natriuretic peptides at inclusion.5. There are important potential confounders that were not evaluated. For instance, there is scarce information about
--

	biomarkers. It appears, natriuretic peptides, sodium, renal function, iron deficiency inflammatory biomarkers were not tested in multivariate analysis. 6. The sample size is small (n=30 at each branch) with many dropouts and it should be stated as an important limitation. 7. The patients included have moderate reduction of exercise capacity and it should be stated. 8. Conclusion: Authors define their study population as a well-characterized cohort of patients with HFpEF. It should be modified. It is important to remark that an external validation of present findings to the whole spectrum of advanced HFrEF (NYHA III-IV) is questionable because the population included in this study exhibit greater functional capacity than observed in daily clinical practice.
--	--

REVIEWER	H M Dalal University of Exeter Medical School Knowledge Spa Royal Cornwall Hospital Truro TR1 3HD
REVIEW RETURNED	29-Mar-2017

GENERAL COMMENTS	The investigators of this single centre (the authors should clarify this in the text) pilot RCT have done well to recruit 60 patients with moderate to severe chronic heart failure to test the feasibility of Low Frequency Electrical Muscle Stimulation. In the abstract (on page 2, line 34-35) the conclusion states that 'A larger RCT in the advanced heart failure population is therefore feasible.' I am not convinced as the text in the concluding line (on page 17, line 15) contradicts this. The conclusion of the abstract should reflect the findings which demonstrate that overall recruitment of 35% does not meet the recommended success criteria (page 5, line 143). The follow up at 20 weeks (Fig 1, page 10) was poor and the authors may wish to expand on how they might improve this in a full RCT given that patients with NYHA III/IV have higher rates of morbidity and mortality. Table 1 (on page 8) would benefit by specifying the severity of CKD e.g. proportion of patients with an eGFR <60 and including the number of patients with implantable cardiac devices such as ICDs The discussion section (on page 15) begins by comparing the poor uptake of the intervention in the trial with poor uptake of CR in the UK. This is not a fair comparison as recruitment and retention in trials of CR are generally better than in the ordinary population. The authors may wish to refer to a recent study by Reeves et al 'A Novel Rehabilitation Intervention for Older Patients with Acute Decompensated Heart Failure- The REHAB-HF Pilot Study' JACC HF 2017) This pilot study also recruited from a patients with heart failure and multiple morbidities. It may prudent to refer to this study as there are several similarities in the participants and what can be learnt from a pilot RCT. It would be useful to have some idea of the proposed sample size for a full RCT given that the authors state in the introduction that the
---

	findings of the pilot 'can be used for sample size calculations' (page 3, line 85) The findings of this pilot RCT are of interest in informing future research that undertaking a full RCT would be challenging. The authors have acknowledged acknowledge this but should make it clear in the abstract in light of the relatively minor improvements in their outcomes. Again, it would be worth looking at the effect sizes reported by Reeves et al (see above) and how they have proceeded to a full RCT.
--	--

VERSION 1 – AUTHOR RESPONSE

Reviewer: 1

Reviewer Name: Patricia Palau

Institution and Country: Hospital General Universitario de Castellón, Cardiology Department, Universitat Jaume I, Spain

Competing Interests: None declared

In this randomized study authors evaluated the feasibility and effectivity of LF-EMS in patients with heart failure with reduced left ventricular ejection fraction (HFpEF).

Authors concluded this study was feasible however it did not show any improvement in functional or QoL in patients with advanced HFrEF.

The hypothesis is pertinent and the protocol was meticulous. Authors should be congratulated. The main strength of this finding is that patients with advanced HFrEF can tolerate a home-based LF-EMS program. However, there are some issues that deserve to be commented/clarified.

Thank you, Dr Palau, for your comments on the manuscript. You have drawn our attention to some key issues that we can hopefully respond to in the revised manuscript and/or in the responses below:

1. Study population: It is necessary to specify the inclusion and exclusion criteria according to recent guidelines.

The recently updated ESC guidelines (2016)(2) on heart failure diagnosis introduced the new definition 'Heart failure with mid-range Ejection Fraction (HFmrEF) for those with EF between 40-50% and stipulates cutoff values for natriuretic peptides to rule out borderline patients. Our inclusion criteria clearly state patients were recruited based on their clinical diagnosis (made based on BNP), NYHA status and echocardiographically documented poor ejection fraction (LVEF<40%)

2. Oeing CU, Tschöpe C, Pieske B. [The new ESC Guidelines for acute and chronic heart failure 2016]. Herz. 2016;41(8):655-63.

2. An important limitation is that heart failure diagnosis was not based on natriuretic peptides

See below response

3. Authors should specify the associated comorbidities, echocardiographic findings and baseline natriuretic peptides in Table 1

The diagnosis of heart failure for participants included the use of NT Pro BNP and the cutoff value of >125 pg/ml was satisfied for all participants. We have amended the manuscript to include baseline BNP values considering the recent ESC guidelines which gives confirmation of the severity of our

study population. As mentioned in response to the editorial comments, the majority of participants did not have ejection fraction measured by Simpsons biplane due to poor echogenicity and for many, persistent arrhythmia, i.e. fast AF. Instead, eyeball assessments of severe LV dysfunction were provided by an experienced echocardiographer and confirmed by the cardiologist in clinic. A note to this effect has been added to the results section (page 8, table 1).

4. Authors selected 60 symptomatic HFrEF (median age 66 years), overweight (BMI 27.8-30.1) with impaired functional capacity by cardiopulmonary exercise testing (CPX) due to shortness of breath or fatigue and NYHA III-IV. However, this is not congruent with the results showed in table 1 where the distance of 6-MWT is 237-337 m. Likewise, previous studies in obese subjects have shown that these individuals display per se a significant impairment of functional capacity. Authors should clarify how many patients exhibit NYHA class II- III –IV and the natriuretic peptides at inclusion.

BNP at baseline has been included based on this recommendation and should strengthen our study. We have also added a comment in the discussion (page 16, line 373) regarding the high variability (caused by a few outliers) around 6mwt which may explain the higher than expected mean 6mwd at baseline. However, previous studies (3, 4) report <300m as prognostic cutoff in heart failure patients associated with an advanced disease state and so we don't feel that this detracts from our assertion that the study population were representative of the advanced heart failure population.

3. Guazzi M, Dickstein K, Vicenzi M, Arena R. Six-minute walk test and cardiopulmonary exercise testing in patients with chronic heart failure: a comparative analysis on clinical and prognostic insights. *Circ Heart Fail.* 2009;2(6):549-55.

4. Pollentier B, Irons SL, Benedetto CM, Dibenedetto AM, Loton D, Seyler RD, et al. Examination of the six minute walk test to determine functional capacity in people with chronic heart failure: a systematic review. *Cardiopulm Phys Ther J.* 2010;21(1):13-21.

5. There are important potential confounders that were not evaluated. For instance, there is scarce information about biomarkers. It appears, natriuretic peptides, sodium, renal function, iron deficiency inflammatory biomarkers were not tested in multivariate analysis.

Sodium, creatinine, potassium urea and creatine kinase were measured alongside NT-pro BNP as a safety measure and as the results section states (page 11, line 292) there were no readings that were abnormal requiring any medical intervention. The authors accept that these biochemical markers may have confounded our results and have now added baseline NT pro BNP and creatinine levels for clarity (page 8, table 1). However, while it may be appropriate to include these markers in a larger RCT multivariate analysis, they were deemed outside the scope of this feasibility study.

6. The sample size is small (n=30 at each branch) with many dropouts and it should be stated as an important limitation.

We recognize that the study sample size is small and this is now stated as one of the limitations. Dropouts during the intervention were within the acceptable a priori criteria for retention, although we have inserted further suggestions for improving dropout in follow-up in the discussion section (page 15, line 351). The sample is suitable for this kind of feasibility/pilot as recommended by Thabane et al (2010)(5).

5. Thabane L, Ma J, Chu R, Cheng J, Ismaila A, Rios LP, et al. A tutorial on pilot studies: the what, why and how. *BMC Med Res Methodol.* 2010;10:1.

7. The patients included have moderate reduction of exercise capacity and it should be stated.

We have added a comment stating this to the discussion (page 16, line 373). See earlier response above

8. Conclusion: Authors define their study population as a well-characterized cohort of patients with HFpEF. It should be modified. It is important to remark that an external validation of present findings to the whole spectrum of advanced HFrEF (NYHA III-IV) is questionable because the population included in this study exhibit greater functional capacity than observed in daily clinical practice.

Whilst we accept that our baseline functional capacity was slightly higher than expected, we must reiterate that the patients were classified as eligible based on their clinical diagnosis using all clinical information available at the time by an experienced heart failure cardiologist.

In line with your comments however, we have modified the a sentence in the participants section (page 4, line 113) to make this point clear.

Reviewer: 2

Reviewer Name: H M Dalal

Institution and Country: University of Exeter Medical School, Royal Cornwall Hospital, Truro, UK

Competing Interests: None

The investigators of this single centre (the authors should clarify this in the text) pilot RCT have done well to recruit 60 patients with moderate to severe chronic heart failure to test the feasibility of Low Frequency Electrical Muscle Stimulation.

Thank you for your review Dr Dalal. We are grateful for your comments and suggestions which we believe are fair and we hope the revised manuscript incorporates your advice.

In the abstract (on page 2, line 34-35) the conclusion states that 'A larger RCT in the advanced heart failure population is therefore feasible.' I am not convinced as the text in the concluding line (on page 17, line 15) contradicts this. The conclusion of the abstract should reflect the findings which demonstrate that overall recruitment of 35% does not meet the recommended success criteria (page 5, line 143).

We accept that our recruitment criterion was narrowly missed in the discussion section, but do not concede that this would completely prohibit a larger trial. However, we have reworded the conclusion of the abstract to mirror the main conclusion

The follow up at 20 weeks (Fig 1, page 10) was poor and the authors may wish to expand on how they might improve this in a full RCT given that patients with NYHA III/IV have higher rates of morbidity and mortality.

We have added an extra section in the discussion (page 15, line 351) that makes suggestions regarding improving dropout.

Table 1 (on page 8) would benefit by specifying the severity of CKD e.g. proportion of patients with an eGFR <60 and including the number of patients with implantable cardiac devices such as ICDs

Patients with ICDs were excluded. eGFRs on all participants were not available as many heart failure patients do not have these tests clinically as part of standard care. However, we have now amended the demographic table (page 8, table 1), to include baseline creatinine levels which give an indication

of kidney function

The discussion section (on page 15) begins by comparing the poor uptake of the intervention in the trial with poor uptake of CR in the UK. This is not a fair comparison as recruitment and retention in trials of CR are generally better than in the ordinary population. The authors may wish to refer to a recent study by Reeves et al 'A Novel Rehabilitation Intervention for Older Patients with Acute Decompensated Heart Failure- The REHAB-HF Pilot Study' JACC HF 2017)

This pilot study also recruited patients with heart failure and multiple morbidities. It may prudent to refer to this study as there are several similarities in the participants and what can be learnt from a pilot RCT.

We accept the point that clinical trials generally have better retention than clinical practice due to extra resource and attention given to this. However, we believe that the uptake to an intervention such as this gives a valid indication of the potential demand for EMS in the heart failure population were it offered routinely.

Thank you for making us aware of the recent study by Reeves et al. We have referenced this paper (page 16, line 373) which we found helpful guiding these amendments. Nevertheless, it should be noted they excluded patients who they deemed at 'risk of 'non-adherence' unlike the pragmatic nature of our trial.

It would be useful to have some idea of the proposed sample size for a full RCT given that the authors state in the introduction that the findings of the pilot 'can be used for sample size calculations' (page 3, line 85)

We previously included a sample size calculation and figure in an earlier draft of the manuscript. It was removed due to word and figure limit restrictions. We have reinstated a brief section (without the figure) on page 14 although the amended manuscript now exceeds the recommended word limit. We trust that this is acceptable to the editor?

The findings of this pilot RCT are of interest in informing future research that undertaking a full RCT would be challenging. The authors have acknowledged this but should make it clear in the abstract in light of the relatively minor improvements in their outcomes. Again, it would be worth looking at the effect sizes reported by Reeves et al (see above) and how they have proceeded to a full RCT.

We have amended the abstract and conclusion of our study to reflect the finding that a larger RCT whilst possible, may not be worthwhile based on the minor improvements observed. The effect sizes from this study are modest in relation to the Reeves study although as mentioned earlier the patients in the Reeves trial were more carefully selected.

VERSION 2 – REVIEW

REVIEWER	Patricia Palau Hospital General Universitario de Castellón Spain
REVIEW RETURNED	21-May-2017

GENERAL COMMENTS	The authors have not addressed most of the concerns I had with the previous submission. Furthermore, there are methodological issues that need to improve (inclusion criteria, exclusion criteria, functional class, echocardiographic findings, other comorbidities, etc.).
--

REVIEWER	Hasnain Dalal University of Exeter Medical School (Truro Campus) UK
REVIEW RETURNED	08-Jun-2017

GENERAL COMMENTS	The authors have made some of the recommended changes which reflect the challenges of proceeding to a fully powered RCT. It is helpful to have the sample size calculation and the authors will be aware of the difficulties in attempting to recruit nearly 500 patients with severe heart failure. They may wish to tone down their claim about the feasibility of a full RCT given that less than half the patients were able to provide the primary outcome at 20 weeks. The latter could be a factor that limits proceeding to a full trial. The study protocol states that all participants will be interviewed at eight weeks to determine acceptability of the intervention. Qualitative data on patient perceptions of the intervention may help to understand the reasons why the follow up at 20 weeks was <50% and why 13 participants declined to take part. Lessons learned could help improve follow up in a future definitive trial. For some reason there was a mismatch in the authors pointers and the manuscript e.g. I was not able to find line 351 on page 15 referring to an extra section on dropout. Similarly page 16 does not have line 373 in my version of the manuscript pdf! The references after reference 29 seem to be jumbled: the authors need to recheck: Page 17 line 32 : Ref 29 is from original manuscript i.e Jeon et al Refs 30 & 31 have been changed from the original manuscript. Although the new refs relate to the 6m walk test. What was the reason for the change Ref 32 & 33 refer to 6m walk test studies, not EMS! Is this the correct? Ref 34 as it is a different ref to the Fulster et al ref in the original manuscript which was a paper about muscle wasting Ref 35 in the original paper was Rullman et al about modifications of skeletal muscle .The current ref is a RCT by Quittan et al . Any reason for the change?
--

VERSION 2 – AUTHOR RESPONSE

Reviewer: 1

Reviewer Name: Patricia Palau

Institution and Country: Hospital General Universitario de Castellón, Spain

Competing Interests: None

The authors have not addressed most of the concerns I had with the previous submission. Furthermore, there are methodological issues that need to improve (inclusion criteria, exclusion criteria, functional class, echocardiographic findings, other comorbidities, etc.).

We are respectfully disappointed that you do not feel that we addressed your previous concerns. Based on your helpful recommendations we added Pro NT BNP levels and echocardiographic findings to the demographic table. While we understand your concern with the inclusion/exclusion criteria we are sure that you would agree that changing these post-hoc is implausible and inappropriate and would therefore not be a good idea. A main driver for this study was that it was completed in a 'real-world' setting, as such, we did not want to unduly bias the study through rigorous inclusion/exclusion criteria. We have added to the limitations section that this study by its nature is 'real world' and by that definition the patients included would be the same were this intervention ever

to be employed in clinical practice. We hope that you find this acceptable, and understand our approach.

Reviewer: 2

Reviewer Name: Hasnain Dalal

Institution and Country: University of Exeter Medical School (Truro Campus), UK

Competing Interests: None declared

The authors have made some of the recommended changes which reflect the challenges of proceeding to a fully powered RCT. It is helpful to have the sample size calculation and the authors will be aware of the difficulties in attempting to recruit nearly 500 patients with severe heart failure. They may wish to tone down their claim about the feasibility of a full RCT given that less than half the patients were able to provide the primary outcome at 20 weeks. The latter could be a factor that limits proceeding to a full trial.

We thank you for this valuable feedback and have revised the abstract and conclusion accordingly. The manuscript now reflects a negative but nevertheless important finding.

The study protocol states that all participants will be interviewed at eight weeks to determine acceptability of the intervention. Qualitative data on patient perceptions of the intervention may help to understand the reasons why the follow up at 20 weeks was <50% and why 13 participants declined to take part. Lessons learned could help improve follow up in a future definitive trial.

Again we are grateful for this valuable suggestion. We have gone back to our source material and analysed the user questionnaires. We have now included a table in the results section summarizing the feedback and refer to this in the discussion. As per the reviewers suggestion we believe that this adds valuable information to the manuscript.

For some reason there was a mismatch in the authors pointers and the manuscript e.g. I was not able to find line 351 on page 15 referring to an extra section on dropout. Similarly page 16 does not have line 373 in my version of the manuscript pdf!

The references after reference 29 seem to be jumbled: the authors need to recheck:

Page 17 line 32 : Ref 29 is from original manuscript i.e Jeon et al

Refs 30 & 31 have been changed from the original manuscript. Although the new refs relate to the 6m walk test. What was the reason for the change

Ref 32 & 33 refer to 6m walk test studies, not EMS! Is this the correct?

Ref 34 as it is a different ref to the Fulster et al ref in the original manuscript which was a paper about muscle wasting

Ref 35 in the original paper was Rullman et al about modifications of skeletal muscle .The current ref is a RCT by Quittan et al . Any reason for the change?

Apologies for this slip up. The references have been double checked now and are in the right place and should hopefully make sense